# Gray Forecast of Ecosystem Services Value and Its Driving Forces in Karst Areas of China: A Case Study in Guizhou Province, China

**DOI:** 10.3390/ijerph182312404

**Published:** 2021-11-25

**Authors:** Sipei Pan, Jiale Liang, Wanxu Chen, Jiangfeng Li, Ziqi Liu

**Affiliations:** 1School of Public Administration, China University of Geosciences, Wuhan 430074, China; pamsp@cug.edu.cn (S.P.); LJL0715@cug.edu.cn (J.L.); 2School of Geography and Information Engineering, China University of Geosciences, Wuhan 430074, China; ziqi@cug.edu.cn; 3Research Center for Spatial Planning and Human-Environmental System Simulation, China University of Geosciences, Wuhan 430078, China; 4State Key Laboratory of Earth Surface Processes and Resource Ecology, Beijing Normal University, Beijing 100875, China

**Keywords:** ecosystem services value, land use change, gray system, driving forces, karst areas, China

## Abstract

A sound ecosystem is the prerequisite for the sustainable development of human society, and the karst ecosystem is a key component of the global ecosystem, which is essential to human welfare and livelihood. However, there remains a gap in the literature on the changing trend and driving factors of ecosystem services value (*ESV*) in karst areas. In this study, Guizhou Province, a representative region of karst mountainous areas, was taken as a case to bridge the gap. *ESV* in the karst areas was predicted, based on the land use change data in 2009–2018, and the driving mechanisms were explored through the gray correlation analysis method. Results show that a total loss of CNY 21.47 billion *ESV* from 2009 to 2018 is due to the conversion of a total of 22.566% of the land in Guizhou, with forest land as the main cause of *ESV* change. By 2025 and 2030, the areas of garden land, water area, and construction land in Guizhou Province will continue to increase, whereas the areas of cultivated land, forest land, and garden land will decline. The total *ESV* shows a downward trend and will decrease to CNY 218.71 billion by 2030. Gray correlation analysis results illuminate that the total population and tertiary industry proportion are the uppermost, among all the driving factors that affect *ESV* change. The findings in this study have important implications for optimizing and adjusting the land use structure ecological protection and will enrich the literature on *ESV* in ecologically fragile areas.

## 1. Introduction

Ecosystem services (ESs) are increasingly the focus of international attention in the 21st century [1], which represents the benefits that life on earth obtains, directly or indirectly, from nature to maintain its ecosystem functions [2,3]. Nevertheless, human activities have exerted a dramatic impact on the ecosystem, resulting in the degradation of approximately 60% of the Earth’s ESs [4,5]. The karst landform, featured in poor ecological stability and sensitive to ecological disturbance [6], are widely distributed globally [7]. Changes of the ecosystem functions, especially the extensive land degradation and vegetation deterioration in karst areas, have caused severe soil erosion and drought [8]. Studies have shown that there is a significantly positive correlation between incidence of poverty and ESs in karst areas [9]. As one of the most fragile ecosystems in the world [10], karst areas in China account for approximately 15% of the total land areas, making China one of the countries with the largest karst landforms in the world [11]. The ecological foundation of karst areas in China is so fragile that it can easily be damaged. Tremendously degraded plant communities were observed in the karst areas of Southwestern China, owing to geological conditions and severe human disturbances [6]. Once damaged, it is formidable to repair and restore, which makes the protection of ESs in karst areas, particularly difficult but momentous. However, ESs changes in karst areas are dealt with in only a few studies, among which, the development trend and drivers of the change are rarely involved. Therefore, quantifying the ESs changes in karst areas and exploring its driving factors will help with formulating practical ESs management policies towards achieving beneficial environmental outcomes [12].

The concept of ESs, first proposed in 1970, provides a valuable perspective on linking human well-being to ecosystem structures and processes [13]. Methods for quantitatively estimating ESs include material quality assessment, value assessment, and energy assessment [14,15], with value assessment being the most widely used one [16]. Experts and scholars have conducted in-depth research on the ecosystem services value (*ESV*) of different ecosystem types, such as forest land [17], farmland [18], grass land [19], regions [20], and cities [16], and a certain degree of progress has been made. The assessment of *ESV*, at different scales in China, also obtains stunning achievements. Yuan et al. [16] calculated the *ESV* loss in China using spatial analysis under the background of urbanization. He et al. [20] introduced the regional difference coefficient to value the ESs of Awang basin, Yunnan, China, from 2012 to 2018. Kuang et al. [21] conducted the spatiotemporal correlation and regression analysis on *ESV* and socioeconomic development indicators in the Xiangxi Tujia-Miao autonomous region, through remote sensing, geographic information system, and R language method. Yet, these studies predominantly focus on the overall changes of the *ESV* in the study area, rarely extended the analysis to specific geological conditions and geomorphic features [22], most notably the karst areas. Furthermore, research on ESs in China’s karst areas predominantly concentrated on the ecosystem degradation [23], ecosystem vulnerability [24], and trade-offs among ESs [8,25], rather few studies analyzed the changes of ESs attributed to land use structure adjustment and the driving factors behind the changes in the karst areas.

ESs are important tools for optimizing territorial space and promoting planning decision better in line with the concept of ecological civilization, the analysis of which can provide theoretical and technical support for different levels of territorial spatial planning and facilitate the ideological synergy in the „Multi-plan Coordination” [25]. To better serve the formulation of territorial spatial planning and provide scientific support, it is of great significance to predict the future development trend of ESs in karst areas. Land use change is a pervasive but important ecological trend worldwide, which is closely related to human and nature [16]. As a complex artificial ecosystem [10], changes in the structure of land use can impact the structure and functions of the ecosystem, which in turn affects the types and intensity of functions provided by the ecosystem [26]. Studies have shown that land use change is the main form of ecosystem change [27,28] and one of the most important drivers of ESs changes [29]. The relationship between *ESV* and land use change has been extensively studied; however, in western China, where the *ESV* hotspots are concentrated [30], especially in the karst areas, relevant research is quite lacking. Hu et al. [22] adopted the benefit transfer method to estimate the *ESV* changes in China’s karst areas during 1992–2015 and to reveal its spatial heterogeneity along with the sensitivity of *ESV* to land use change. Peng et al. [7] took Guizhou Province as the case study area and distinguished the impact of land use and climate change on ESs in karst landscape in China. Therefore, it is scientifically reasonable and technically feasible to apply land use change data to measure ESs change in karst areas. Land use prediction methods, include GM(1,1) (Gray Model) [31], Markov chain [32], cellular automata [33], CLUE-S model [34,35], and FLUS (Future Land-Use Simulation) model [36], were widely used in existing literature. Compared with other land use prediction methods, gray GM(1,1) can make predictions with a small number of data samples, without considering the distribution pattern of land use structure and the transition probability of land use types, which exhibits unique advantages of convenient calculation and high prediction accuracy [10].

Currently, research concerning the driving factors of *ESV* changes are plentiful [37,38,39,40,41], while direct drivers, such as land use change, and the application of new technologies attracts more attention [42,43]. For example, Liu et al. [44] superimposed the impact of future climate scenarios and land use change on ESs to explore the overall effect of the changes on ESs. During the period of rapid economic development, population growth, economic development, and urban expansion have exerted unprecedented pressures and negative effects on the regional ESs [39], and the impact of national policies on the ESs should not be neglected, such as the Grain for Green Program [17,45,46]. However, studies on whether socioeconomic characteristics are related to ESs prefer conceptual model and scenario analysis [26]. Villamagna et al. [47] insisted that socioeconomic factors are the driver of the ESs in agricultural landscape. Minin et al. [48] explored the trade-offs of ESs, land use, economic development, and biodiversity to identify priority areas through four policy scenarios. Song et al. [41] illuminated that socioeconomic development effect exerted the greatest impact on wetland *ESV* in Northeast China. A preliminary qualitative conclusion can be drawn from the above research, that socioeconomic factors do impose effect on ESs; yet, they cannot distinguish or sort the specific types of socioeconomic factors, which, as a result, cannot facilitate targeted policy application.

Guizhou is one of the provinces with the most typical karst landform development in China, exhibits extensive karst landform features and obvious regional differentiation [10]. In the past decade, Guizhou has experienced rapid economic development and urbanization, which have profoundly affected the karst ecosystem. Conducting research on kasrt areas is a new challenging attempt, which can enrich the types of studies on *ESV* in ecologically fragile areas. Taking Guizhou Province as a study case, this study analyzes the spatio-temporal change of land use and *ESV* during 2009–2018 and its driving forces in Guizhou Province. Based on the gray GM(1,1) model, the study made gray prediction on the future land use structure and the development trend of *ESV*. Additionally, the driving factors behind the change of *ESV* were explored by adopting gray correlation analysis method. The application of gray system to forecast the development trend and reveal the driving forces behind is an innovation. The findings of this research can aid in targeted policy application towards achieving beneficial environmental outcomes. The research objectives are as follows:(1)To analyze the spatio-temporal characteristics of land use change and *ESV* in Guizhou Province from 2009 to 2018.(2)To forecast the development status and trend of *ESV* in Guizhou Province in 2025 and 2030.(3)To explore the driving factors affecting *ESV* change in Guizhou Province.

## 2. Materials and Methods

### 2.1. Study Area

Guizhou Province (103°36’~109°36’ E, 24°37’~29°14’ N) is located in the east slope of Yunnan-Guizhou Plateau, with the terrain high in the west and low in the east (Figure 1). Guizhou Province covers an area of 17,609,858 hm^2^, accounting for 1.83% of the total land area of China. The landform types are complex and diverse, mainly plateau and mountain, exhibiting typical karst geomorphological characteristics. The forest coverage rate is reaching 60%, and the land reclamation rate is as high as 20.03%, which is approximately twice the national average level. The high-intensity reclamation of territorial space has caused large areas of rocky desertification and soil erosion. Under the background of rapid urbanization, industrialization, and significantly unbalanced regional development, the land use in Guizhou Province underwent profound transformation from 2009 to 2018. The area of construction land increased by 181,959 hm^2^, whereas the area of cultivated land decreased by 36,340 hm^2^, the forest land decreased by 84,117 hm^2^, and the grass land decreased by 59,077 hm^2^. Land use change disturbs the ecosystem severely, and Guizhou Province has extensive karst landform features, generating a combination of serious land degradation issues [49]. Therefore, conducting research on the impact of land use change on *ESV* and unearthing the driving factors behind the *ESV* change in Guizhou is of great significance, which can serve as scientific guidance for the ecological protection in karst areas, as well as to promote the coordinated development of ecological protection and social economy.

### 2.2. Data Sources

This study used ten-year data from 2009 to 2018. In details, the land use data are predominantly from the land use change survey data of Guizhou Province, and the socioeconomic data are from China Statistical Yearbook and Guizhou Statistical Yearbook. DEM is a 90 m ASTER GDEM data derived from the Geospatial Data Cloud (http://www.gscloud.cn/, accessed on 15 November 2021), which can accurately reflect the topographic features of the study area. According to China’s land use classification system, the original land use classes of the data consist of seven first-class land use types, including cultivated land, garden land, forest land, grass land, construction land, water area, and unused land.

### 2.3. Method

#### 2.3.1. Evaluation Model of Ecosystem Services Value

Costanza et al. [50] clarified the principles and methods of *ESV* estimation in a scientific sense. In China, Xie et al. [2,51], based on Costanza’s research and China’s characteristics, grouped China’s ecosystem and re-determined the *ESV* coefficient, which became the basis for Chinese scholars to assess *ESV* [10,52,53]. Based on the research results of Costanza et al. [50] and Xie et al. [2,51], the literature on the study of *ESV* in different regions were analyzed, the ecosystem types and ecological value coefficients (*VC*), corresponding to the land use categories in the Guizhou Province, were determined by combining the geographical and climate situation of the study area (Table 1). The calculation of *ESV* is as follows:*ESV* = ∑A_i_ × *VC_i_*,(1)
where *ESV* refers to the total value of ESs in the study area (CNY); *A_i_* refers to the distribution areas (hm^2^) of the *i* type of land use in the study area; *VC_i_* refers to the ecological *VC* per unit area of type *i* land use category (CNY /hm^2^ yr); *i* refers to the type of land use in the study area.

#### 2.3.2. Sensitivity Index

Coefficient of sensitivity (*CS*) or coefficient of elasticity is usually used to determine the sensitivity and robustness of the price (coefficients) [54]. To verify the representative of ecosystem types to each landscape type, as well as the accuracy of *VC*, *CS* was introduced to determine the degree of dependence of the total *ESV* on the change of *VC* over time, namely changes in *ESV* attribute to a 1% change in *VC*, which can be expressed in Equation (2) [55]:(2)$$CS=|\frac{\left(ESV_{j}-ESV_{i}\right)/ESV}{\left(VC_{jk}-VC_{ik}\right)/VC_{ik}}|,$$
where *i* and *j* represent the initial and adjusted value, respectively; *k* represents the land use category. The *CS* is applied to verify whether the selected *VC* is suitable for this study area.

According to Equation (2), *CS* is always less than 1 [20]. The higher the ratio, the more critical the accuracy of the *VC*, and the greater the sensitivity of the *ESV* to it. In this study, the *VC* of each landscape type was adjusted up and down by 50%, respectively, to illustrate the sensitivity of *ESV* to *VC*.

#### 2.3.3. Gray Forecast Model

Gray prediction model [10,31] is based on the understanding of the uncertain characteristics of the system evolution, using sequence operators to generate and process the original data, dig out the law of system evolution, establish the gray model, and make scientific quantitative predictions for the future state of the system. In this study, the gray GM(1,1) model is adopted to forecast the land use structure and the development trend of *ESV* in Guizhou Province in 2025 and 2030.

If $x_{t}^{\left(0\right)}\left(i\right)$ is defined as the original data sequence and $x_{t}^{\left(1\right)}\left(i\right)$ is a sequence of number generated by an accumulation, then the standard equation of the first-order, linear constant coefficient differential equation of GM(1,1) model is as follows:(3)$$\frac{dx^{\left(1\right)}}{dt}+ax^{\left(1\right)}=u,$$

The standard solution, corresponding to the gray GM(1,1) model, is as follows:(4)$$x^{\left(1\right)}=\left(x^{\left(0\right)}\left(1\right)-\frac{u}{a}\right)e^{-at}+\frac{u}{a},$$
where *a* and *u* are unknown parameters to be determined; *t* is time.

To improve the accuracy and reliability of gray GM(1,1) model prediction, a posterior difference test method was adopted to test the model accuracy. The posterior error ratio C and the small error frequency P are defined as follows:(5)$$C=\frac{S_{2}}{S_{1}},p=P\{\left|\varepsilon_{k-}\bar{\varepsilon }\right|<0.6745S_{1}\text{\}},$$
where *S_1_* is the standard deviation of the original data; *S_2_* is the mean of standard deviation of the predicted data; $\varepsilon_{k}$ is the error of forecast data; and $\bar{\varepsilon }$ is the prediction error. The value of *C* indicates the degree of dispersion of the difference between the predicted value and the actual value (the lower, the better).

#### 2.3.4. Gray Correlation Analysis

Gray correlation analysis is a method to measure the correlation degree between factors according to the similarity or difference degree of the development trend among factors. Its principle is based on the macro or micro geometric approximation between behavior factors [56]. When the experiment is not explicit, or the experimental method cannot be carried out accurately, gray analysis can help to make up for the deficiencies in statistical regression [57] and can be used to measure the approximate correlation between the series [58]. The more similar the curves, the greater the correlation between relative data series [56]. In other words, if the changing trends of two factors (or indicators) are basically consistent or synchronized, it can be regarded as the correlation between the two is relatively strong, and vice versa. In this study, seven factors were selected from three aspects of population, economy, and policy to analyze the driving factors of *ESV* in Guizhou Province, which were the total population (TP), urbanization level (UL), gross domestic product (GDP), secondary industries proportion (SIP), tertiary industries proportion (TIP), total investment in fixed assets (TIFA), and afforestation project (AP) (Table 2).

After the original data are normalized by the initial value or average value, the gray correlation coefficient can be obtained by the following equation:(6)$$\xi_{\mathrm{ij}}\left(k\right)=\frac{\underset{i}{\min}\underset{k}{\min}\left|x_{0}\left(k\right)-x_{i}\left(k\right)\right|+\rho \underset{i}{\max}\underset{k}{\max}\left|x_{0}\left(k\right)-x_{i}\left(k\right)\right|}{\left|x_{0}\left(k\right)-x_{i}\left(k\right)\right|+\rho \underset{i}{\max}\underset{k}{\max}\left|x_{0}\left(k\right)-x_{i}\left(k\right)\right|},$$
where *ξ_ij_(k)* is called the gray correlation coefficient at time *t = k* for x_0_ and x_i_, $\underset{i}{\min}\underset{k}{\min}\left|x_{0}\left(k\right)-x_{i}\left(k\right)\right|$ and $\underset{i}{\max}\underset{k}{\max}\left|x_{0}\left(k\right)-x_{i}\left(k\right)\right|$ are the minimum difference and maximum difference from the normalized reference series to all the other normalized series, respectively, and *ρ* is called the distinguished coefficient which can be adjusted to better distinguish between the normalized reference series and normalized comparative series, *ρ* is taken between 0 and 1, usually 0.5.

Then, Equation (7) is introduced to calculate the gray correlation grades, which is actually the average of the gray correlation coefficients.
(7)$$r_{ij}=\frac{1}{n}\sum_{k=1}^{n}\xi_{i}\left(k\right),$$
where *r_ij_* is the gray correlation grade. *n* refers to the number of factors. The closer to 1 the existing gray correlation grade is, the greater the impact of the comparison sequence on the reference series, that is, the greater the correlation impact on the *ESV*.

## 3. Results

### 3.1. Land Use Change in Guizhou Province

As shown in Table 3, whether in 2009 or 2018, forest land accounted for more than 50% of the total land areas, constituting the main body of the land use structure. Due to the large areas of forest land, the ecological function in Guizhou is of great significance and it is sensitive in the construction of ecological environment. From 2009 to 2018, the total area of land use change in Guizhou Province was 374,900 hm^2^. Among them construction land and forest land are the main types of land use change, with a total change area of 266,100 hm^2^ (construction land increased by 182,000 hm^2^, forest land decreased by 84,100 hm^2^), accounting for 70.98% of the total land use change. Among the land use categories, the area of forest land, grass land, cultivated land, and unused land all decreased to varying degrees (Table 3), with a reduction of 84,100 hm^2^, 59,100 hm^2^, 36,300 hm^2^, and 8400 hm^2^, respectively. The reduction rates were 0.93%, 3.6%, 0.8%, and 1.7%, respectively. Construction land, garden land and water area increased by 182,000 hm^2^, 4700 hm^2^, and 1300 hm^2^, respectively, with an increase rate of 12.0%, 3.0%, and 0.5%. The gray forecast analysis shows that by 2025 and 2030, cultivated land, forest land, grass land, and unused land exhibit a downward trend, whereas garden land, construction land, and water areas will be reversely varied (Figure 2).

The main driving factors for the decrease of cultivated land and the increase of garden land are the implementation of the Grain for Green Program [17,45,46] and the adjustment of the agricultural industry structure, which has turned part of the cultivated land into orchards, tea gardens, and other land use types. The decrease in forest land, grass land and unused land, and the increase in construction land is predominantly due to the development of social economy [10]. Additionally, Guizhou Province has implemented a large number of resettlements, infrastructure, poverty alleviation industries, and people’s livelihood security projects, and the demand for construction land is far beyond expectations. The reason for the increase of water area is that the three-year battle of water transport construction launched by Guizhou Province promoted the construction of water conservancy and hydropower projects.

### 3.2. Ecosystem Services Value in Guizhou Province

#### 3.2.1. Change of Total Ecosystem Services Value in Guizhou Province

The total *ESV* in Guizhou Province from 2009 to 2018 is calculated by Equation (1) (Table 4). Results show that the total *ESV* in Guizhou Province is CNY 223.64 billion in 2009 and CNY 221.50 billion in 2018, a decrease of CNY 21.47 billion in 10 years, with a change rate of 9.60%. The *VC* of forest land is relatively high, leading to the continuous decrease of total *ESV* in the Guizhou Province, due to the reduction of forest land in the past decade. In ten years, the area of forest land decreased by 84,117 hm^2^, resulting in a reduction of CNY 16.26 billion in *ESV*, accounting for 75.73% of the total *ESV* decrease, indicating that the change of forest land is the main reason for the change of *ESV*. In terms of the total *ESV* of each land use type, the *ESV* of cultivated land, forest land, grass land, and unused land exhibit a downward trend, whereas garden land and water area display a floating change (Figure 3). The main reason for the decline in *ESV* in Guizhou is predominantly, due to the development of social economy. Though the implementation of the Grain for Green Program did play an active role [10,59,60], leading to the rapid growth of garden land, but it only worked in the early stage. The three-year battle of water transport construction in Guizhou exerted great impact on the change of water area, but then the effect weakened. The decrease of *ESV* caused by construction occupation, due to economic development is difficult to compensate. Besides, the total *ESV* of forest land and cultivated land reach up to 89%. Therefore, Guizhou Province should attach importance to the protection of forest land and cultivated land ecosystem in the future land use, continue to strengthen the ecological environment construction through forest land protection and comprehensive land management.

Since the *ESV* distribution characteristics of different regions vary little each year, we only listed the maps in 2018 (Figure 4). The *ESV* provided by cultivated, forest, and grass land were predominantly distributed in areas of the Wumeng Mountains, Dalou Mountains, and south of the Miaoling Mountains (Figure 4a,c,d), while the *ESV* provided by the garden land is predominantly distributed in south of the Miaoling Mountain and the north of the Wuling Mountain (Figure 4b). The *ESV* provided by water area and unused land is predominantly distributed in the central and western part of Guizhou Province (Figure 4e,f).

#### 3.2.2. Sensitivity Analysis

The *CS* can determine the dependence of *ESV* over time on ecological *VC*, so as to verify whether the selected ecological *VC* is suitable for this study area. By adjusting the *VC* by ±50%, the *ESV* and *CS* of each category in Guizhou Province in 2009 and 2018 were calculated (Table 5). The horizontal comparison shows that the ecosystem *CS* is in the order of forest land > cultivated land > grass land > water area > garden land > unused land, and the highest value is 0.779. Namely, when the *VC* of forest land increases by 1%, the corresponding *ESV* increases by 0.779%, exerting the greatest impact on the *ESV* of the study area. The lowest value is 0.001, that is, when the *VC* of the garden or unused land increases by 1%, the corresponding *ESV* increases by 0–0.001%, which has little impact on the *ESV* of the study area. From the perspective of different periods, there is no significant change in the sensitivity index of all land types from 2009 to 2018. The greatest change is forest land, but it only increases by 0.007%. In addition, the 10-year change rate of the total *ESV* in the study area varies from 0.897% to 1.563% after the adjustment of ecological *VC*, and the difference is small. All these indicate that *ESV* in the study area is inelastic to *VC*, namely the *VC* adopted in the study area does not affect the authenticity of *ESV* changes over time, which is in line with the actual situation in the study area, and the research results are credible.

Since the spatial distribution pattern of the sensitivity index of *ESV* to *VC* of different land use types were similar in different years, we only listed the maps in 2018 (Figure 5). The spatial distribution showed that the sensitivity index of cultivated land in the west was higher than that in the east (Figure 5a), whereas the sensitivity index of forest land shows the opposite (Figure 5c). Besides, the sensitivity index of garden land, grass land, water area, and unused land show higher distribution characteristics in the middle and southwest Guizhou Province (Figure 5b,d–f).

### 3.3. Gray Forecast of Ecosystem Services Value

Based on the land use data of Guizhou Province from 2009 to 2018, Matlab_R2021a was applied to construct a gray GM(1,1) model suitable for all land use types in the study area to predict the land use structure in 2025 and 2030. The forecast assumes that the total area of the study area remains unchanged, and the predicted value of unused land is the total area of Guizhou Province minus the predicted value of other land use types and the *ESV* of different land use types is calculated (Table 6). Results show that by 2025 and 2030, the *ESV* of garden land and water area will increase slightly, whereas others will decrease to varying degrees, with forest land the greatest (Figure 6). The total *ESV* will decrease to CNY 219.83 billion and CNY 218.71 billion in 2025 and 2030, respectively. According to the model prediction accuracy rating standard, when C < 0.35 and *p* ≥ 0.95, the model accuracy is level 1 (good). When C ≥ 0.65 and *p* ≤ 0.75, the model accuracy is level 4 (unqualified). According to Table 6, the posterior error ratio of this model, Ci ≤ 0.28 < 0.35, and Pi = 1 > 0.95, shows that the prediction accuracy of the model is good and the prediction value is highly reliable.

### 3.4. Driving Forces of Change in Ecosystem Services Value in Guizhou Province

Gray correlation grades of influential factors on *ESV* change are listed in Table 7. It can be seen that all the gray correlation grades are greater than 0.5 and maintain high value, indicating that the seven parameters are the driving factors affecting the *ESV* change in Guizhou Province. From 2009 to 2018, the relative importance of influencing factors to *ESV* change is as followed: TP > TIP > SIP > AP > UL > GDP > TIFA. Among them, TP exerts the greatest impact on *ESV*, with a gray correlation grade of 0.9965, which is close to 1. The smallest gray correlation grade belongs to TIFA, which equals to 0.6620. Except from 2009 to 2012 TIP accounts for the highest gray correlation degree, TP ranks the highest of gray correlation degree in other years, which has been maintained above 0.97. Besides, the gray correlation grade of TIP has also remained above 0.93, revealing that TP and TIP were the two main driving factors of *ESV* change in Guizhou Province.

## 4. Discussion

### 4.1. Comparing with Previous Studies

Judging from the overall change trend of *ESV* in Guizhou Province, the ecological protection trend in karst areas is not optimistic. Studies have shown that the process of urbanization in the past is often at the cost of the ecological environment, resulting in a significant loss of *ESV* [26]. In the early stage (2000–2008), the implementation of Grain For Green Program did play an active role in ESs in karst areas [10,59,60], yet, according to our study, in the past decade, though the Grain For Green Program continued, the land demand resulted from urbanization and population boom far exceeded expectations, leading to the decline of *ESV* in karst areas. From 2009–2018, construction land in Guizhou Province has increased by 181,959 hm^2^, with a change rate of 11.96%. As a result, *ESV* in Guizhou Province has decreased by CNY 2.15 billion, with an average annual decrease of 3.6%. Undoubtedly, urbanization has exerted a negative impact on the ecosystem supply capacity, which is in line with the research of Chen et al. [61].

In this study, *ESV* was evaluated in combination with the equivalent value table proposed by Xie et al. [2,51] and Costanza et al. [50]. Results showed that the contribution of different land use types to the total *ESV* was significantly different, among which the forest land contributes the largest, followed by cultivated land and grass land, which is consistent with the research of Leh et al. [62] and W. Chen et al. [63]. Some studies believe that the *ESV* of construction land is negative [64], whereas some studies consider that construction land can provide certain ESs (e.g., entertainment, tourism, culture, and carbon storage) [22,65]. This research is in conformity to the research of Wu et al. [26], which suggested that construction land contributes nothing to *ESV* but the loss of it; hence, we assign the coefficient of construction land to be 0.

Previous studies mostly used coarse data sets that lacked correction, leading to differences in *ESV* assessment in the same place [66,67]. Meanwhile, the inconsistency in the land use classification can also lead to differences in *ESV* assessment. For example, when Wang (2018) [68] measured the *ESV* of Guizhou from 2016 to 2017, the land use types were divided into cultivated land, forest land, water area, grass land, construction land, and unused land; the calculated *ESV* for 2016 and 2017 were CNY 207.90 and CNY 207.70 billion, respectively, both less than the *ESV* calculated in this study, which are CNY 221.87 and CNY 221.66 billion, respectively. Zhao et al. [69] classified land use types into five categories: cropland, forest land, grass land, water bodies, and unused land. According to their calculation results, the total amount of *ESV* in 2013 and 2017 were both higher than that in this study, CNY 261.80 and CNY 261.20 billion, respectively. In the evolution of China’s land use classification system, garden land is an important secondary category. However, when *ESV* is measured, garden land is rarely considered, but mostly counted into forest land [70], which is also a factor attracts the difference in *ESV* assessment. This study believes that garden land differs from forest land, the *VC* of which should be less than that of forest land. Therefore, it is listed separately for evaluation.

### 4.2. Evolution Mechanism of Ecosystem Services Value in Guizhou Province

Our research results are in accordance with previous studies that land use change is one of the main drivers of *ESV* loss [71], and the most significant change in ESs attributed to land use change is predominantly urbanization driven [7]. However, the research results also show that neither socioeconomic factors nor physical factors are the sole or primary potential causes of ESs change [27], and the main driving forces are also variable in different periods [72]. In addition to land use structure, economic, social, demographic, and policy factors are also driving factors affect *ESV*. The results of this study show that from 2009 to 2018, TP imposes the greatest impact on *ESV*, followed by TIP in economic factors. Leading driving forces vary in different periods. The most important factor affecting the regional variation of *ESV* is TIP, between 2009 and 2012, while TP between 2012 and 2018. The urbanization process is accompanied by the population surge, and the rise of the tertiary industry has become an important symbol of the acceleration of urbanization. The Guizhou Province is located in the center of the east Asian karst region, with the most complex, complete types, and largest distribution area in the world [70]. Thus, the analysis of the driving factors affecting the *ESV* change in Guizhou will provide decision-making reference for the study of *ESV* change and ecological governance in karst areas and will enrich the literature on *ESV* in ecologically fragile areas.

### 4.3. Policy Implications

In the early stage of economic development, ecological considerations have commonly been interpreted as a factor restricting economic development and can, thus, be ignored easily [12]. Existing evidence indicates that economic development yields obvious ecosystem degradation through land use change [27]. Rapid urbanization in Guizhou has driven the expansion of human-dependent lands (e.g., housing and transportation infrastructure), while ecological lands (e.g., forest land, grass land, and garden land) were heavily encroached. The ratio of GDP to *ESV* in China is, approximately, 1:1 [73], whereas the ratio of GDP to *ESV* in Guizhou from 2009 to 2018 increased from 1.75:1 to 6.93:1, indicating that the *ESV* in Guizhou is far below the average level in China. Policy can play a vital role in maintaining regional *ESV* [74,75]. A considerable number of studies have focused on ESs, and how to incorporate ESs into land use and ecological conservation decision-making has long been discussed [76]. However, the non-uniqueness and effectiveness of assessment methods along with the difficulty of incorporating assessment results into actual management have led to the policy implications of ESs [77]. Additionally, the trade-offs and synergies in ESs [78], are also the key to land use decision-making [42]. For example, land development to guarantee food security will inevitably lead to the weakening of water and soil conservation and the hydrological regulation function.

As a typical representative of China’s karst landform, Guizhou is positioned as a province with balanced grain production, marketing, energy, and mineral base. Meanwhile, it is also as an important ecological security barrier in the upper reaches of the Yangtze river and the Pearl River. Thus, its ecological restoration should be placed in a prominent position; under the premise of protection, the economic development must meet the carrying capacity of resources and environment. Within this context, Guizhou Province must coordinate ecological protection and economic development. In accordance with the principle of „ecological priority and green development” [61], the adjustment of economic structure should be accelerated, industrial transformation and upgrading must be promoted, and rationalization of the spatial layout of industrial functions must be guided, so that to promote green and high-quality development. Meanwhile, it would also be an innovation to incorporate new concepts, such as green GDP, into the performance evaluation system for government officials [3,79].

### 4.4. Limitations and Future Directions

Certain limitations could be observed in this study. The *VC* adopted in this study is based on the research of Xie et al. [2,51] and Costanza et al. [50]. Actually, the *VC* of the same ESs in different regions have certain deviations. However, this study does not consider applying different indicators to estimate the same *ESV* in different regions. Besides, obtaining data in the karst areas is not easy, which can be a direction of future research. Although the assessment of ESs has been extensively studied, considerable progress still needs to be made to improve the evaluation of ESs [80]. In predicting future *ESV* changes, the temporal heterogeneity of *VC* is also easy to be ignored. Therefore, how to accurately assess and quantify ESs in a complex and constantly changing ecosystem will also be a question worth exploring. Furthermore, the research of Aschonitis et al. [54] showed that the ecosystem sensitivity index can only be applied to rank the importance of land use based on its contribution to the total *ESV* and should be abandoned for assessing the robustness and sensitivity of ESs. Future studies can be discussed in this regard. In terms of land use simulation and prediction, the gray prediction model has the advantage of being able to make predictions, in the case of a small number of data samples; however, the impact of factors, such as policies, in the long-term series will make for the prediction results to be inconsistent with the actual development situation, which needs to be noted when applied for forecasting in the future. In the next step, our study will further improve the method model and analyze the value differences of ecosystem types in different times within the same region, and the spatial distribution value differences of ecosystem types will be further explored.

## 5. Conclusions

The gray GM(1,1) model is introduced in this study to reveal the spatial-temporal changes of land use and *ESV*. The contribution of different land use types to *ESV*, in virtue of the sensitivity index, is measured. The gray correlation analysis method is adopted to explore the driving mechanism affect *ESV* changes in Guizhou Province. The areas of forest land, grass land, cultivated land, and unused land in Guizhou Province decreased to varying degrees, whereas the construction land, garden land, and water areas exhibited an upward trend from 2009 to 2018. Among them, construction land and forest land are the main types of land use change. The *ESV* in Guizhou Province decreased by CNY 21.47 billion, with forest land the main cause of *ESV* change. The sensitivity indexes of *ESV* to *VC* were all less than l, indicating that the evaluation results of *ESV* in Guizhou Province were credible. From the perspective of different periods, there was no significant change in the sensitivity index of all land use types from 2009 to 2018. The forecast results showed that by 2025 and 2030, the areas of garden land, water area, and construction land in Guizhou Province will continue to increase, whereas the areas of cultivated, forest and garden land will be on a decline. Nevertheless, the total *ESV* will decrease to CNY 218.71 billion by 2030. Gray correlation analysis results showed that population, economy, and policy factors can all affect *ESV* change, among which TP and TIP are the two most important driving factors. Consequently, top priority should be given to the ecological factors in future planning and socio-economic development. We expect that the research result and proposed policy suggestions of this study will constitute an important aid to the correct treatment of the spatial characteristics of land uses change and *ESV* in karst areas.

## Figures and Tables

**Figure 1 ijerph-18-12404-f001:**
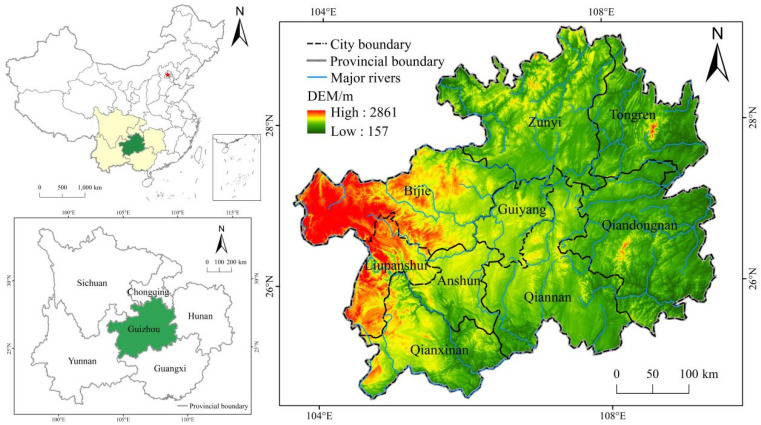
Geographical location of Guizhou Province.

**Figure 2 ijerph-18-12404-f002:**
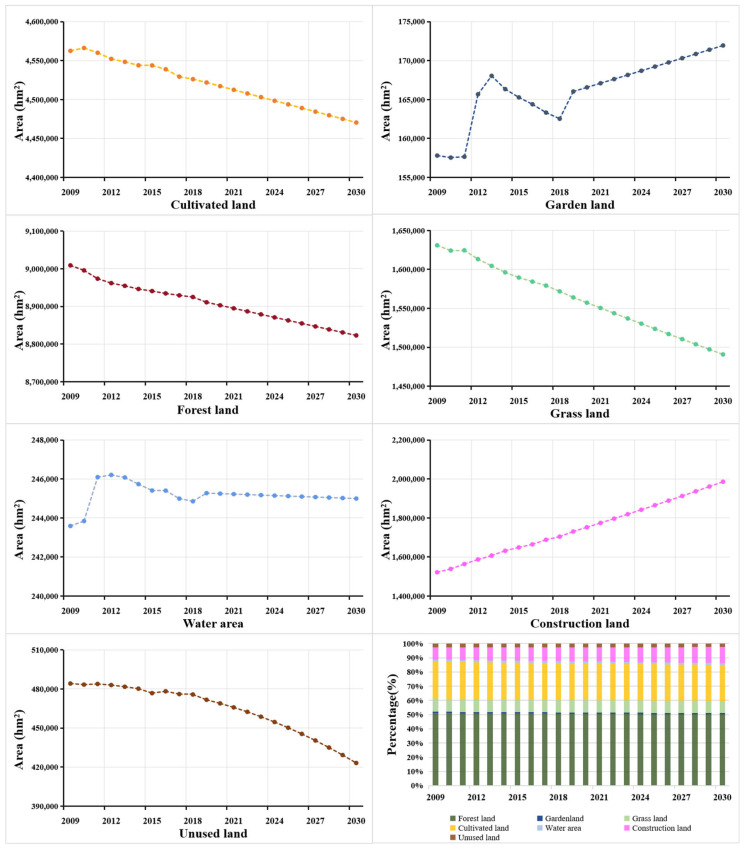
Land use area change of Guizhou Province from 2009 to 2030.

**Figure 3 ijerph-18-12404-f003:**
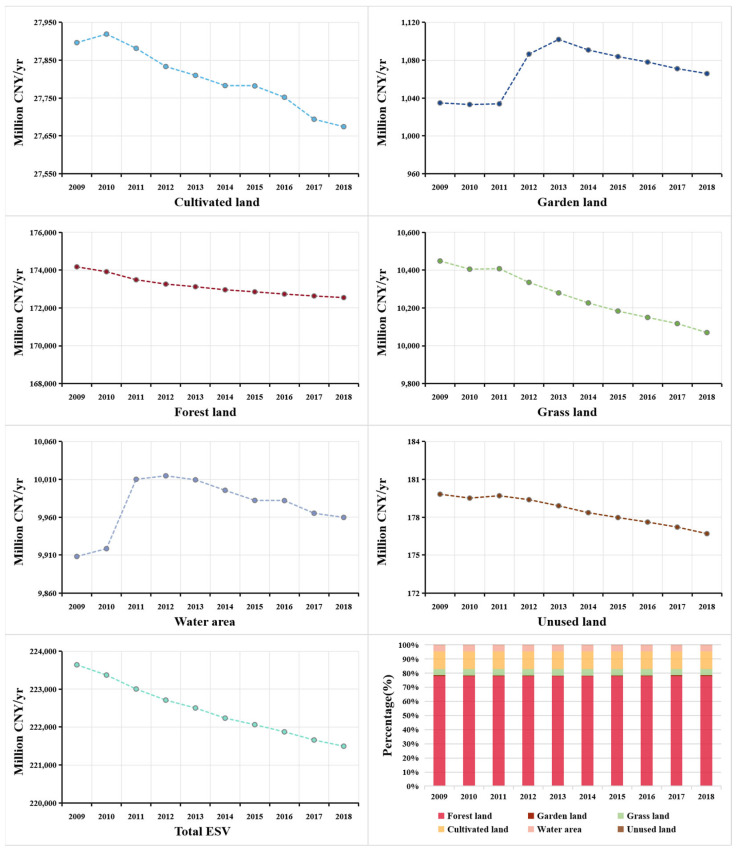
Change of ecosystem services value of different land use types in Guizhou Province from 2009 to 2018.

**Figure 4 ijerph-18-12404-f004:**
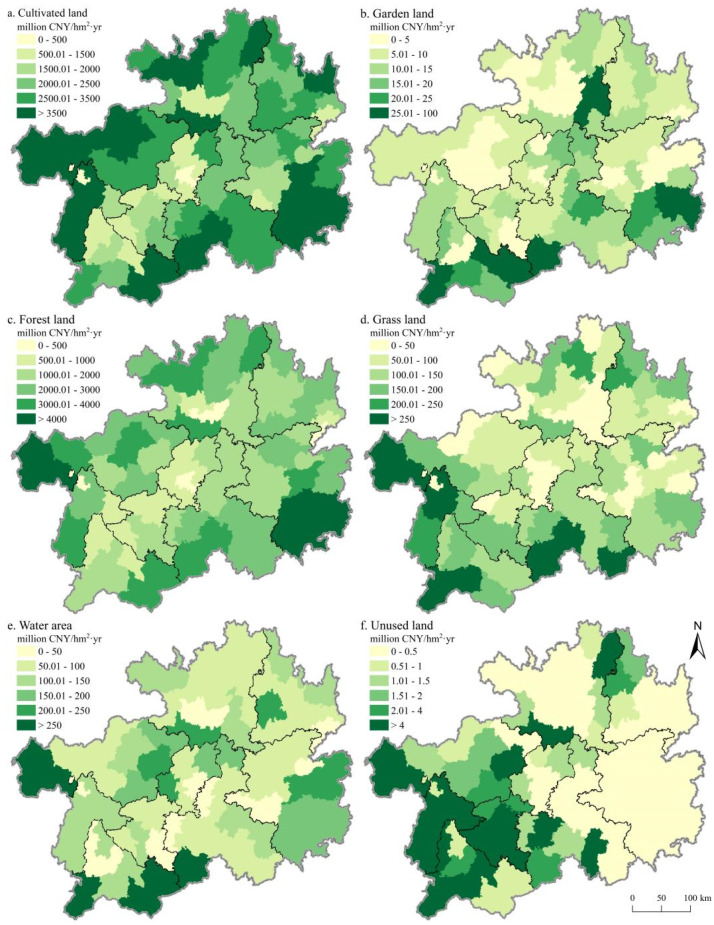
Spatial distribution characteristics of ecosystem services value of different land use types in Guizhou in 2018. Notes: (**a**) the spatial distribution characteristics of ecosystem services value of cultivated land; (**b**) the spatial distribution characteristics of ecosystem services value of garden land; (**c**) the spatial distribution characteristics of ecosystem services value of forest land; (**d**) the spatial distribution characteristics of ecosystem services value of grass land; (**e**) the spatial distribution characteristics of ecosystem services value of water area; and (**f**) the spatial distribution characteristics of ecosystem services value of the unused land.

**Figure 5 ijerph-18-12404-f005:**
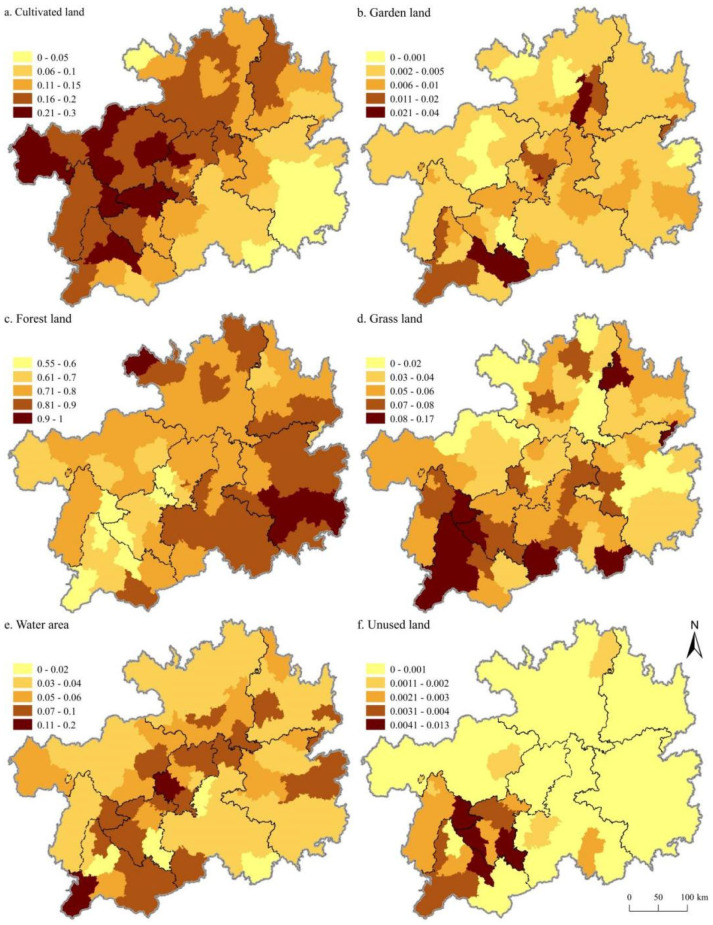
Spatial pattern of sensitivity coefficients of different land use types in Guizhou Province in 2018. Notes: (**a**) the spatial pattern of the sensitivity coefficient of cultivated land; (**b**) the spatial pattern of the sensitivity coefficient of garden land; (**c**) the spatial pattern of the sensitivity coefficient of forest land; (**d**) the spatial pattern of the sensitivity coefficient of grass land; (**e**) the spatial pattern of the sensitivity coefficient of water area; (**f**) the spatial pattern of the sensitivity coefficient of unused land.

**Figure 6 ijerph-18-12404-f006:**
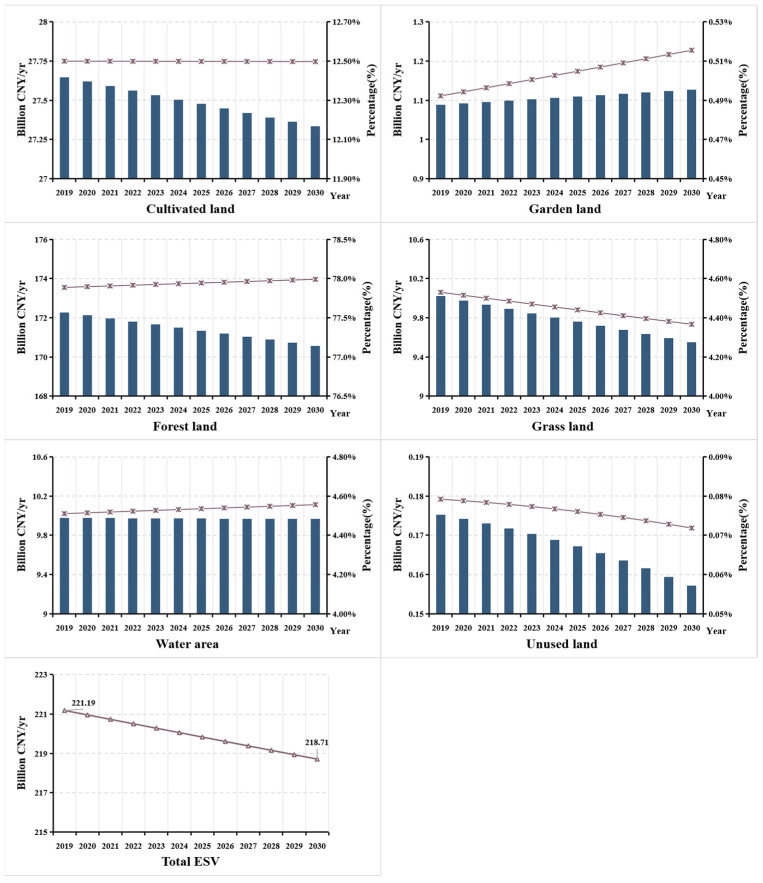
Forecast changes of ecosystem services value of different land use types in Guizhou Province from 2019 to 2030.

**Table 1 ijerph-18-12404-t001:** Ecological value coefficients of different land use categories in the study area [2,10,50,51].

ESs		Land Use Category/(Million CNY/hm^2^ yr)					
		Forest Land	Garden Land	Grass Land	Cultivated Land	Water Area	Unused Land
Regulating services	Gas regulation	3097.00	1265.50	707.90	442.40	0	0
	Climate regulation	2389.10	1170.30	796.40	787.50	407.00	0
	Hydrological regulation	2831.50	41.50	707.90	530.90	18,033.20	26.50
	Waste treatment	1159.20	722.10	1159.20	1451.20	16,086.60	8.80
Supporting services	Soil formation and retention	3450.90	1291.90	1725.50	1291.90	8.80	17.70
	Biodiversity protection	2884.60	16.60	964.50	628.20	2203.30	300.80
Supplying services	Food production	88.50	356.90	265.50	884.90	88.50	8.80
	Raw material	2300.06	1145.40	44.20	88.50	8.80	0
Cultural services	Recreation and culture	1132.60	547.80	35.40	8.80	3840.20	8.80
Total		19,333.46	6558.00	6406.50	6114.30	40,676.40	371.40

**Table 2 ijerph-18-12404-t002:** Variables used to identify the driving forces of ecosystem services value change in Guizhou Province.

Category	Variable	Description	Source	Mean, Range
Population	TP	Total population at the year-end	Guizhou Province Bureau of Statistics	35.24, 34.69 to 36 million
	UL	Urbanization level	Guizhou Province Bureau of Statistics	0.39, 0.30 to 0.48 percent
Economic	GDP	Gross domestic product	Guizhou Province Bureau of Statistics	896.34, 391.27 to 1535.32 billion CNY
	SIP	Secondary industry proportion	Guizhou Province Bureau of Statistics	38.50, 35.86 to 41.6 percent
	TIP	Tertiary industry proportion	Guizhou Province Bureau of Statistics	47.76, 44.6 to 50.09 percent
	TIFA	Total investment in fixed assets	Guizhou Province Bureau of Statistics	903.60, 245.10 to 1836.36 billion CNY
Policy	AP	Afforestation project	Guizhou Province Bureau of Statistics	25,149.27, 18,053.33 to 50,586.67 hm^2^

**Table 3 ijerph-18-12404-t003:** Changes of the land use area in Guizhou Province in 2009 and 2018.

Land Use Type	Area of 2009 /hm^2^	Proportion /%	Area of 2018 /hm^2^	Proportion /%	Variation /hm^2^	Rate of Change /%
Cultivated land	4,562,515	25.909	4,526,175	25.703	−36,340	−0.796
Garden land	157,805	0.896	162,535	0.923	4730	2.997
Forest land	9,008,978	51.159	8,924,861	50.681	−84,117	−0.934
Grass land	1,630,923	9.261	1,571,846	8.926	−59,077	−3.622
Construction land	1,521,870	8.642	1,703,828	9.675	181,959	11.956
Water area	243,584	1.383	244,853	1.390	1269	0.521
Unused land	484,183	2.749	475,759	2.702	−8423	−1.740
Total	17,609,858	100	17,609,858	100	--	--

**Table 4 ijerph-18-12404-t004:** Changes of the total ecosystem services value in Guizhou Province in 2009 and 2018.

	*ESV*					
	2009		2018		Variation /Billion CNY	Rate of Change /%
	Value /Billion CNY	Proportion /%	Value /Billion CNY	Proportion /%		
Cultivated land	27.90	12.474	27.67	12.494	−2.22	−0.796
Garden land	1.04	0.463	1.07	0.481	0.31	2.997
Forest land	174.18	77.881	172.55	77.902	−16.26	−0.934
Grass land	10.45	4.672	10.07	4.546	−3.79	−3.622
Water area	9.91	4.430	9.96	4.497	0.52	0.521
Unused land	0.18	0.080	0.18	0.080	−0.03	−1.740
Construction land	0	0	0	0	0	0
Total	223.64	100	221.50	100	−21.47	−9.600

**Table 5 ijerph-18-12404-t005:** Estimation of the coefficient of sensitivity and ecosystem services value after the adjustment of the total ecosystem services value coefficient in Guizhou Province.

	*ESV*/Billion CNY				The Difference of the Rate of Change before *VC* Is Not Adjusted/%	*CS*	
	2009	2018	Variation	Rate of Change /%		2009	2018
Cultivated land *VC* + 50%	237.59	235.33	−2.26	−0.951	−0.951	0.125	0.126
Cultivated land *VC*−50%	209.69	207.66	−2.04	−0.971	−0.971		
Garden land *VC* + 50%	224.16	222.03	−2.13	−0.951	−0.951	0.005	0.005
Garden land *VC* − 50%	223.13	220.96	−2.16	−0.969	−0.969		
Forest land *VC* + 50%	310.73	307.77	−2.96	−0.953	−0.953	0.779	0.786
Forest land *VC* − 50%	137.37	135.22	−2.15	−1.563	−1.563		
Grass land *VC* + 50%	228.68	226.53	−2.15	−0.939	−0.939	0.047	0.045
Grass land *VC* − 50%	218.42	216.46	−1.96	−0.897	−0.897		
Water area *VC* + 50%	228.60	226.48	−2.12	−0.928	−0.928	0.044	0.045
Water area *VC* − 50%	218.69	216.52	−2.17	−0.994	−0.994		
Unused land *VC* + 50%	223.73	221.58	−2.15	−0.961	−0.961	0.001	0.001
Unused land *VC* − 50%	223.55	221.41	−2.15	−0.960	−0.960		

**Table 6 ijerph-18-12404-t006:** Ecosystem services value in Guizhou Province in 2025 and 2030.

	2025		2030		Accuracy Test	
	Predictive Value/hm^2^	*ESV*/Billion	Predictive Value/hm^2^	*ESV*/Billion	P	C
Cultivated land	4,493,780	27.48	44,705	27.33	1	0.15
Garden land	169,230	1.110	1719	1.13	1	0.28
Forest land	8,862,768	171.35	88,229	170.58	1	0.20
Grass land	1,523,707	9.76	14,909	9.55	1	0.11
Water area	245,119	9.97	2450	9.97	1	0.16
Unused land	450,216	0.17	4502,	0.16	-	-
Construction land	1,865,038	0	19,856	0	1	0.06
Total	17,609,858	219.83	17,609,858	218.71	-	-

**Table 7 ijerph-18-12404-t007:** Gray correlation grades of influential factors on ecosystem services value change.

Driving Forces		Gray Correlation Grades (R)			
		2009–2018	2009–2012	2012–2015	2015–2018
Population	TP	0.9965	0.9845	0.9957	0.9748
	UL	0.9140	0.8491	0.9558	0.8695
Economic	GDP	0.7909	0.7130	0.7314	0.7215
	SIP	0.9883	0.9632	0.5082	0.9458
	TIP	0.9921	0.9873	0.9830	0.9318
	TIFA	0.6220	0.6251	0.8189	0.6442
Policy	AP	0.9379	0.8441	0.9369	0.7018

## Data Availability

The data that support the findings of this study are available from the corresponding author upon reasonable request.

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
