# Peer review of "Gray Forecast of Ecosystem Services Value and Its Driving Forces in Karst Areas of China: A Case Study in Guizhou Province, China"

_ijerph, 2021, doi:10.3390/ijerph182312404_

Round 1
Reviewer 1 Report
Please find attached.

Reviewer 2 Report
I have carefully considered and read the manuscript entitled “Gray Forecast of Ecosystem Services Value and its Driving Forces in Karst Areas of China: A case study in Guizhou Province,China” and have the following observations:
1) It is recommended not to repeat keywords from the title but to include others not mentioned in the title, in order to increase the visibility of the article.
2) Please highlight your contribution and novelty of this manuscript with accuracy in the introduction part.
3) The acronyms should be defined at first appearance in the manuscript and then must be consistently used throughout the manuscript. Some acronym is missing.
4) Explain RBM meaning and GM meaning.
5) km2 should be replaced by km2 (2.1. Study area)
6) Equation number 6 is missing or the correlative number from 7 has been lost.
7) If possible in the text and in the tables the units should be homogeneous, km2, hm2, ha. Table 6 km2, table 1, hm2, Figure 2 ha. Should be redacted tables.
8) I suggest the author(s) further discuss table 4 and better discuss table 5.
9) Reading the figures in tables and text is very difficult. It should present the data better. The use of the decimal separator or grouped figures is very important. There are major errors with the use and absence of the decimal separator in the tables and in the text. Agreement is required in the text when using two or three digits. Presentation needs to be improved.
10) Recheck the references and their style are according to the journal requirements, and in-text and end-text should be the same and vice versa. The bibliography should be changed in all manuscript, and also the references style. Is very incorrect presentation of the bibliography.
11) Review the bibliography in case there is one more recent
12) I recommend completing the conclusions with a more personal contribution on the situation that has arisen and possible strategies future
13) At the end of the paper, when the authors mention some of its limitations, and possible future lines of research, they should indicate if they are prepared to address them…And in this case how to solve the limitations.
Reviewer 3 Report
This manuscript is well organized to predict the ecosystem services value in Guizhou Province. The findings are important and interesting. But authors used published results to calculate ESV in study area, which generates uncertainties, because these published paper did not focus on the Guizhou Province. Different study area had different climate, soil type, hydrology and vegetation types, and the ESV must be calculated based on the local environment conditions. Thus, it is important that the published data can represent the reasonable ESV in Guizhou, please clarify in the revised version.
